# Therapeutic doses of ketamine acutely attenuate the aversive effect of losses during decision-making

Mariann Oemisch[1], Hyojung Seo[2]*

[1]Department of Neuroscience, Yale School of Medicine, New Haven, United States; [2]Department of Psychiatry and Neuroscience, Yale School of Medicine, New Haven, United States

*For correspondence:
Hyojung.seo@yale.edu

Competing interest: The authors declare that no competing interests exist.

**Abstract** The discovery of rapid-acting antidepressant, ketamine has opened a pathway to a new generation of treatments for depression, and inspired neuroscientific investigation based on a new perspective that non-adaptive changes in the intrinsic excitatory and inhibitory circuitry might underlie the pathophysiology of depression. Nevertheless, it still remains largely unknown how the hypothesized molecular and synaptic levels of changes in the circuitry might mediate behavioral and neuropsychological changes underlying depression, and how ketamine might restore adaptive behavior. Here, we used computational models to analyze behavioral changes induced by therapeutic doses of ketamine, while rhesus macaques were iteratively making decisions based on gains and losses of tokens. When administered intramuscularly or intranasally, ketamine reduced the aversiveness of undesirable outcomes such as losses of tokens without significantly affecting the evaluation of gains, behavioral perseveration, motivation, and other cognitive aspects of learning such as temporal credit assignment and time scales of choice and outcome memory. Ketamine's potentially antidepressant effect was separable from other side effects such as fixation errors, which unlike outcome evaluation, was readily countered with strong motivation to avoid errors. We discuss how the acute effect of ketamine to reduce the initial impact of negative events could potentially mediate longer-term antidepressant effects through mitigating the cumulative effect of those events produced by slowly decaying memory, and how the disruption-resistant affective memory might pose challenges in treating depression. Our study also invites future investigations on ketamine's antidepressant action over diverse mood states and with affective events exerting their impacts at diverse time scales.

## eLife assessment

The authors use reinforcement learning modeling to study the alterations following acute ketamine in macaques. The evidence supporting the conclusion that ketamine reduces the impact of losses vs. neutral/gains is **solid**. In this version of this **valuable** study, the authors make more measured interpretations about the relationship between the processing of losses and ketamine's antidepressant effects.

## Introduction

Depression is a debilitating mental illness that causes enormous personal sufferings as well as societal and economic burden (*Whiteford et al., 2013*; *Greenberg et al., 2021*; *Mrazek et al., 2014*). The discovery of ketamine, a non-competitive *N*-methyl-D-aspartate (NMDA) receptor antagonist as a rapid-acting antidepressant, inspired extensive research to understand the mechanisms of ketamine's

action as well as a new perspective on the neurobiology of depression that depression might reflect non-adaptive changes in the glutamatergic and GABA (gamma aminobutyric acid)-ergic signal transmission of the prefrontal and limbic system in response to environmental stress and adversity (*Berman et al., 2000*; *Zarate et al., 2006*; *Sanacora et al., 2008*; *Lapidus et al., 2014*; *Krystal et al., 2013*; *Abdallah et al., 2015*; *Duman et al., 2016*; *Murrough et al., 2017*; *Krystal et al., 2019*; *Duman et al., 2019*). Despite recent progress in understanding the mechanisms of ketamine's antidepressant action at cellular, synaptic, and network levels, many questions remain regarding enhancing the specificity to target depressive symptoms without undesirable side effects as well as long-term safety for repeated dosing (*Krystal et al., 1994*; *Sanacora et al., 2017*; *Short et al., 2018*; *Bonaventura et al., 2021*).

A major gap in our knowledge that hampers the effort to design novel treatments with high specificity and low off-target effect lies at the functional-systems level to link cellular and synaptic phenomena to physiological and behavioral processes that directly cause the symptoms of depression. In light of symptomatic heterogeneity within the diagnostic category of mental disorders including depression, the importance of integrative and comprehensive analyses with regard to multiple functional domains/systems has been recognized to promote interpretable and translatable results from pre-clinical and clinical research (*Cuthbert, 2014*; *Kozak and Cuthbert, 2016*). Nevertheless, this type of analyses and results for assessing the behavioral effect of ketamine's antidepressant action have seldom been available from the behavioral tests prevalently used in rodents (e.g. forced swim test, sucrose preference test, etc.) or self-reports from human subjects/patients. Another type of scarce data is early behavioral signs of ketamine's action that might mediate and/or predict its longer-term effects. Ketamine has shown to induce rapid-onset (<1 hr) cellular, molecular, physiological, and connectivity changes that are thought to reflect and/or mediate its antidepressant action (*Li et al., 2010*; *Yang et al., 2018*; *Al Shweiki et al., 2020*; *Grimm et al., 2015*; *Scheidegger et al., 2016*; *Gould et al., 2019*). In contrast, the acute behavioral effect that can be indicative of and/or predict longer-term effects has been hard to isolate from dissociative side effects.

Here, we assessed the effects of therapeutic doses of ketamine (0.5–1 mg/kg) administered intramuscularly (IM) or intranasally (IN) on cognitive, affective, and motivational aspects of learning and decision-making. Three rhesus monkeys performed a token-based biased matching pennies (BMP) game against a computerized opponent in which they gained or lost tokens as the outcome of their decisions (*Seo and Lee, 2009*; *Seo et al., 2014*). We found that ketamine acutely reduced the adverse effect of undesirable outcomes such as losses of tokens, which was independent of other side effects such as ocular nystagmus. However, ketamine did not significantly change animals' motivation to earn reward, behavioral perseveration, or other cognitive aspects of learning and memory. These results constitute a rare report of an acute effect of therapeutic dose of ketamine on the processing of affectively negative events during dynamic decision-making. Our study also demonstrates the sensitivity and the potential of the non-human primate model in assessing behavioral and systems-level mechanisms of antidepressant action as well as investigating the pathophysiology of depression when combined with recently developed tools for large-scale neural recording and circuit manipulation.

## Results

Three rhesus monkeys were trained to play a token-based BMP game against a computerized opponent, in which their choice typically depended on the outcomes obtained from each choice in the past trials (*Figure 1A, B*). In the control sessions where saline was administered, animals tended to repeat the choice that yielded a token, while switching away from the choice that led to non-gain outcomes, with a stronger tendency to switch after loss than neutral outcome (i.e. zero token) (*Equation 1*; *Figure 1C*). The reinforcing and punishing effects of gain and non-gain outcomes also decayed exponentially over time, with the recent outcomes exerting a larger effect than remote ones. Ketamine selectively reduced the punishing effect of non-gain outcomes, with significantly larger change being induced after loss than neutral outcome (*Figure 1C*). In addition, the ketamine-induced modulation was larger for the effect of recent than remote outcomes (analysis of variance [ANOVA], three-way interaction among outcome, trial lag, and drug condition, $F_{(8, 1964)} = 4.05$, $F_{(8, 1649)} = 4.46$, and $F_{(8, 509)} = 3.62$ for animal P, Y, and B, respectively. $p < 0.001$ in all animals for IM administration of 0.5 mg/kg; $F_{(8, 1229)} = 4.89$, $p < 0.001$ in animal P; $F_{(8, 389)} = 1.89$, $p = 0.06$ in Y; $F_{(8, 1002)} = 3.27$, $p = 0.001$ in B for IN administration of 1 mg/kg). 0.25 mg/kg of IM-administered and 0.5 mg/kg of

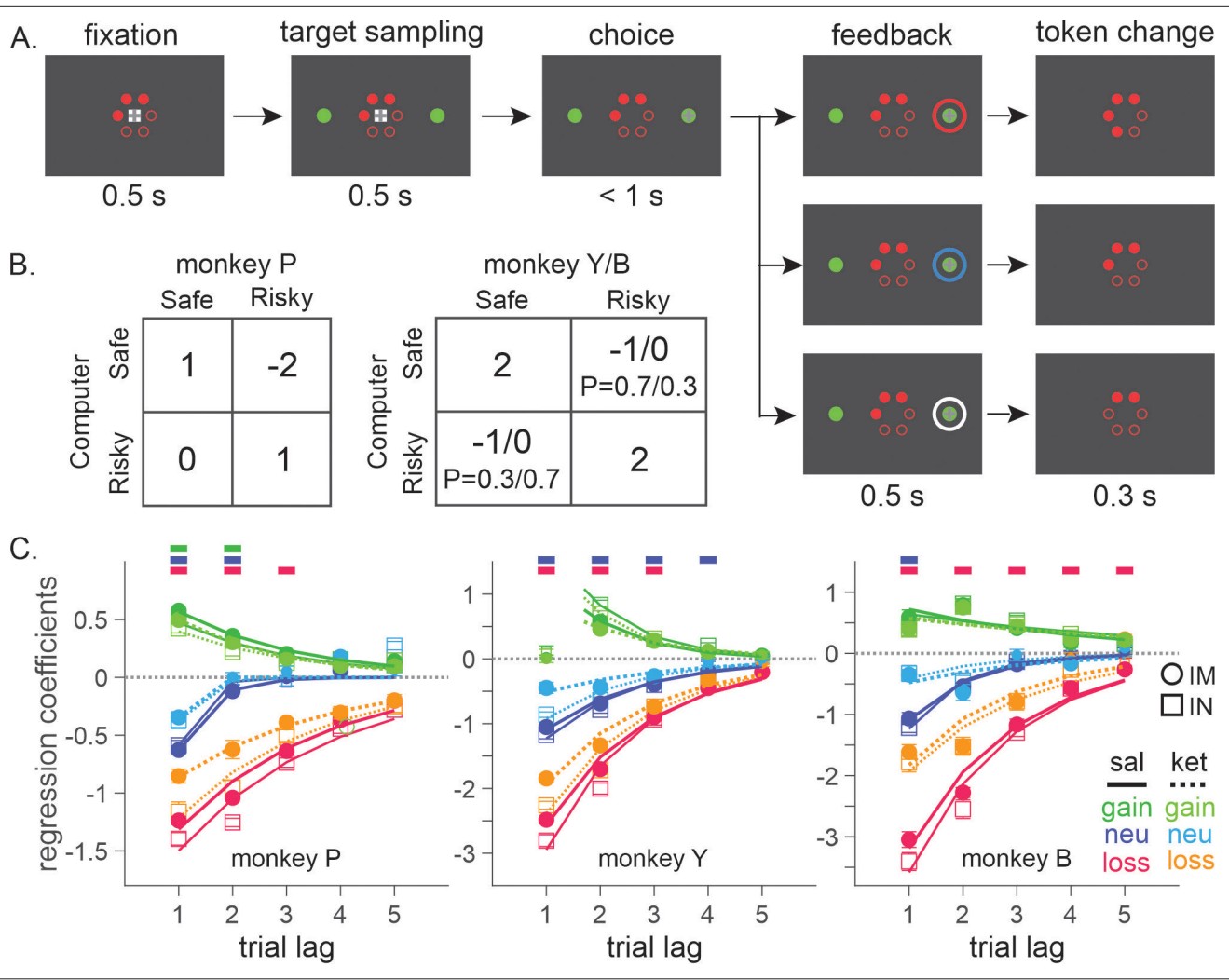

**Figure 1.** Biased matching pennies (BMP) task and ketamine-induced behavioral modulation. (**A**) Temporal sequence of trial events. Gray cross indicates the target that the animal was required to fixate during each epoch. Each trial began with the animal's gaze on the fixation target displayed on the center of a computer monitor for 0.5 s. Solid red disks around the fixation target indicated the tokens owned by the animal, namely asset, with empty disks serving as placeholders for the tokens to be acquired for exchange with juice reward. After two green disks were displayed for 0.5 s at the diametrically opposed positions along the horizontal meridian, the fixation target was extinguished signaling the animal to indicate its choice by shifting gaze to one of the two green disks. After 0.5 s of fixation, a feedback ring appeared around the chosen target with its color indicating the choice outcome, followed by the corresponding change in tokens. Once the animal collected six tokens, they were automatically exchanged with six drops of apple juice. After juice reward, the animal began the subsequent trial with two to four free tokens. Tokens and placeholders stayed on the screen throughout the trial and inter-trial interval. (**B**) Payoff matrix of BMP. (**C**) Coefficients from logistic regression models applied separately to the data from saline (sal) and ketamine (ket) sessions with intramuscular (IM, filled symbols) and intranasal (IN, empty symbols) administration. Standard errors are shown as horizontal bars above/below each coefficient. Lines are exponential functions best fit to the coefficients from saline (solid) and ketamine (dotted) sessions, and those from IM (thick) and IN (thin) sessions. Bars at the top of each panel indicate that the difference between saline and ketamine sessions is statistically significant with the horizontal position and color of each bar coding trial lag and outcome (same color scheme as that of saline sessions), respectively (independent sample $t$-test, $p < 0.05$; IM and IN sessions were combined). For IM sessions, N=52 (Sal) and 17 (Ket) for monkey P; N=49 (Sal) and 15 (Ket) for monkey Y; N=25 (Sal) and 9 (Ket) for monkey B. For IN sessions, N=23 (Sal) and 13 (Ket) for monkey P; N=16 (Sal) and 10 (Ket) for monkey Y; N=19 (Sal) and 8 (Ket) for monkey B.

The online version of this article includes the following source data and figure supplement(s) for figure 1:

**Source data 1.** Means and standard errors of regression coefficients, and best-fit parameters of exponential functions.

**Figure supplement 1.** Behavioral effect of 0.25 mg/kg intramuscularly (IM) administered (top) and 0.5 mg/kg intranasally (IN) administered (bottom) ketamine.

**Figure supplement 1—source data 1.** Means and standard errors of regression coefficients.

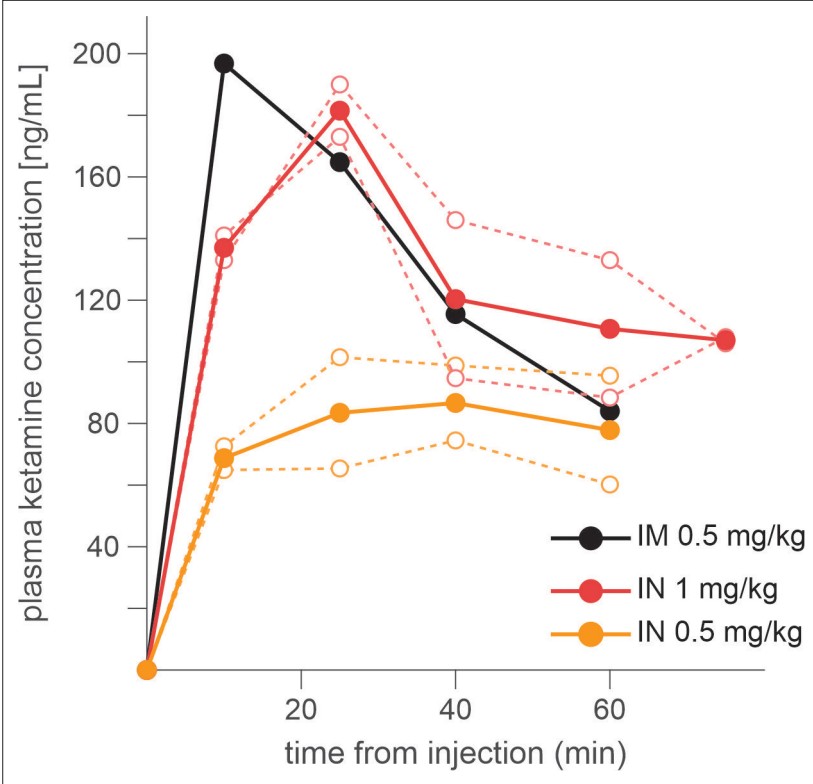

**Figure 2.** Time course of plasma concentration of ketamine following intramuscular (IM) and intranasal (IN) administration. Dotted lines indicate the data from individual sessions, and dotted lines for average across individual sessions. Blood sample was taken every 20 min after injection.

The online version of this article includes the following source data for figure 2:

**Source data 1.** Numerical data for plasma concentration used to generate the figure.

IN-administered ketamine produced qualitatively similar results (*Figure 1—figure supplement 1*). We focused our main analyses on the data collected with 0.5 mg/kg IM- and 1 mg/kg IN-administered ketamine, which produced comparable plasma concentration and behavioral effects (*Figure 2*). Plasma concentration and its time course over 60 min were also comparable to those measured after 0.5 mg/kg in human subjects (*Zarate et al., 2012*).

Outcome-dependent choice behavior and its modulation by ketamine were also parametrically modeled by exponential function, in which the initial impact of each outcome at the time of delivery gradually decayed over subsequent trials (*Equation 2*; *Figure 1C*). Ketamine significantly reduced initial amplitude/impact of non-gain outcomes ($b_0^k$) consistently in all three animals, and with both IM ($b_0^k$ = 0.45 (P), 0.63 (Y), 1.36 (B) for loss, *t*-test, p < $10^{-11}$ for all animals; $b_0^k$ = 0.28 (P), 0.58 (Y), 0.61 (B) for neutral outcome, p < $10^{-5}$ for all animals) and IN administration ($b_0^k$ = 0.29 (P), 0.56 (Y), 1.36 (B) for loss, p < $10^{-11}$ for all animals; $b_0^k$ = 0.22 (P), 0.31 (Y), 0.83 (B) for neutral outcome, p < 0.001 for all animals). Ketamine did not significantly modulate the time constant of decay, and the modulation of initial impact was sufficient to explain the behavioral phenomena.

Dependence of choice on the cumulative effect of temporally discounted past outcomes is the signature of reinforcement learning (RL) (*Lee et al., 2012*). We used RL models to gain further insights into the possible mechanisms underlying ketamine-induced behavioral modulation (see Materials and methods). We first determined the variant/class of RL model that best explained the animals' choice behavior under normal conditions (i.e. saline sessions) (*Table 1*). The differential forgetting (DF) model best fit the data in all three animals, in which the value functions of chosen and unchosen action decayed at different rates, with forgetting being slower for chosen than unchosen action (*Figure 3* and Table 3).

**Table 1.** Reinforcement learning models for normal behavior during saline sessions.

| Model | Value update |
|---|---|
| Q-learning (Q) | $Q_{t+1}(A) = \alpha_F \cdot Q_t(A) + (1 - \alpha_F) \cdot R_t$ , if A is chosen. <br> Rt = 1, 0, −1 for gain, neutral, and loss, respectively. |
| Q-learning with subjective outcome evaluation (QSE) | $Q_{t+1}(A) = \alpha_F \cdot Q_t(A) + \Delta_t$ , if A is chosen. <br> $\Delta_t = \Delta_G$ , $\Delta_N$ , $\Delta_L$ for gain, neutral, and loss (QSE). <br> $\Delta_N = 0$ (QSE-R) |
| Differential forgetting (DF) | $Q_{t+1}(A) = \alpha_{F-C} \cdot Q_t(A) + \Delta_t$ , if A is chosen. <br> $Q_{t+1}(A) = \alpha_{F-UC} \cdot Q_t(A)$ , if A is not chosen. <br> $\Delta_t = \Delta_G$ , $\Delta_N$ , $\Delta_L$ for gain, neutral, and loss (DF). <br> $\Delta_N = 0$ (DF-R) |
| Non-differential forgetting (NDF) | $Q_{t+1}(A) = \alpha_F \cdot Q_t(A) + \Delta_t$ , if A is chosen. <br> $Q_{t+1}(A) = \alpha_F \cdot Q_t(A)$ , if A is not chosen. <br> $\Delta_t = \Delta_G$ , $\Delta_N$ , $\Delta_L$ for gain, neutral, and loss (NDF). <br> $\Delta_N = 0$ (NDF-R) |
| DF with asset-gated outcome evaluation (DF-A) | $Q_{t+1}(A) = \alpha_{F-C} \cdot Q_t(A) + \lambda \cdot asset_t \cdot R_t$ , if A is chosen. <br> $Q_{t+1}(A) = \alpha_{F-UC} \cdot Q_t(A)$ , if A is not chosen. <br> $R_t$ = 1, 0, −1 for gain, neutral, and loss, respectively. |

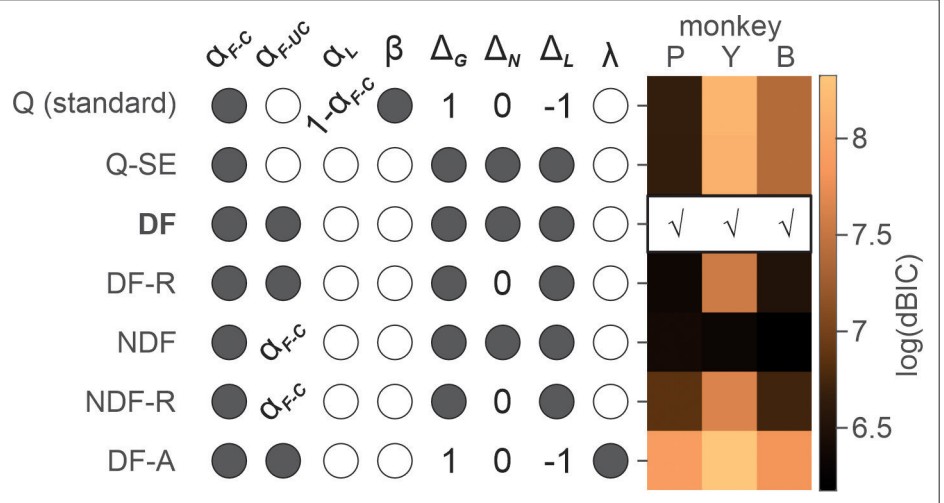

**Figure 3.** Model comparison for saline sessions. Each row shows free (filled circles), unused (empty circles), and fixed (specific values) parameters of a variant of reinforcement learning (RL) model. For the details of RL models and parameters, see *Table 1*. Heat map on the right represents natural logarithm of differential BIC (Bayesian information criterion) of each variant from the best model. Best model for each animal is indicated by check (√) mark. *Q(standard)*: standard Q-learning model; *Q-SE*: Q-learning with subjective outcome evaluation; *DF*: differential forgetting model; *DF-R*: differential forgetting with neutral outcome being the reference point; *NDF*: non-differential forgetting model; *NDF-R*: non-differential forgetting model with neutral outcome being the reference point; *NDF-A*: non-differential forgetting model with asset-gated outcome evaluation.

The online version of this article includes the following source data for figure 3:

**Source data 1.** Numerical values of BIC used to generate the figure.

**Table 2.** Reinforcement learning models for ketamine-induced modulation of choice behavior.

| K-model | Hypothesis | Model parameters |
|---|---|---|
| | **Value model** | |
| 1 | Common modulation of LR/value for non-gain outcomes | $\Delta_t = \Delta_N^S + k$, $\Delta_L^S + k$, for neutral and loss outcomes, respectively. |
| 2 | Differential modulation of LR/value for non-gain outcome | $\Delta_t = \Delta_N^S + k_1$, $\Delta_L^S + k_2$, for neutral and loss outcomes, respectively. |
| 3 | Differential modulation of LR/value for gain vs. non-gain outcome | $\Delta_t = \Delta_G^S + k_1$, $\Delta_N^S + k_2$, $\Delta_L^S + k_2$, for gain, neutral, and loss outcomes. |
| 4 | Differential modulation of LR/value for all the outcomes | $\Delta_t = \Delta_G^K$, $\Delta_N^K$, $\Delta_L^K$, for gain, neutral, and loss outcomes, respectively. |
| | **Memory model** | |
| 5 | Differential modulation of forgetting rate for chosen and unchosen actions | $\alpha_{F-C}^K$, $\alpha_{F-UC}^K$ for chosen and unchosen action, respectively. |
| | **Perseveration model** | |
| 6 | Modulation of perseveration tendency | $X_t^K = (1 - \varepsilon) \cdot [Q_t(R) - Q_t(L)] + \varepsilon \cdot I_t^C$ <br> $I_t^C = 1\ (-1)$, if right (left) target was chosen at trial $t$. |
| 7 | Modulation of LR/value and perseveration | Model 4 + model 6 |
| | **Temporal credit assignment (TCA) model** | |
| 8 | Increase mis-assignment | $X_t^K = [Q_t(R) - Q_t(L)] + \omega_1 \cdot I_{t-2}^c \cdot \Delta_{t-1}$ <br> $+ \omega_2 \cdot I_{t-3}^c \cdot \Delta_{t-1} + \omega_3 \cdot I_{t-4}^c \cdot \Delta_{t-1}$ <br> $I_t^C = 1\ (-1)$, if right (left) target was chosen at trial $t$. <br> $\Delta_t^s = \Delta_G^S$, $\Delta_N^S$, $\Delta_L^S$ for gain, neutral, and loss t trial $t$. |
| 9 | Increase spread | $X_t^K = [Q_t(R) - Q_t(L)] + \omega_1 \cdot I_{t-1}^c \cdot \Delta_{t-2}$ <br> $+ \omega_2 \cdot I_{t-1}^c \cdot \Delta_{t-3} + \omega_3 \cdot I_{t-1}^c \cdot \Delta_{t-4}$ <br> $I_t^C = 1\ (-1)$, if right (left) target was chosen at trial $t$. <br> $\Delta_t^s = \Delta_G^S$, $\Delta_N^S$, $\Delta_L^S$ for gain, neutral, and loss at trial $t$. |
| 10 | Increase statistical learning (SL) | $X_t^K = [Q_t(R) - Q_t(L)] + \omega_1$ <br> $\cdot I(|rFreq_t| > \tau) \cdot rFreq_t \cdot avg\Delta_t$ <br> $rFreq_t = Freq_{R,t} - Freq_{L,t}$ <br> $,Freq_{A,t} = \rho * Freq_{A,t-1} + I_{chosen}$ <br> $I(|rFreq_t| \geq \tau) = 1\ (0)$, if $|rFreq_t| \geq (<) \tau$ <br> $avg\Delta_t = \varphi * avg\Delta_{t-1} + \Delta_t^S$ <br> $\Delta_t^s = \Delta_G^S$, $\Delta_N^S$, $\Delta_L^S$ for gain, neutral, and loss at trial $t$. |
| | **Motivation model** | |
| 11 | Modulation of asset-gated outcome evaluation | $Q_{t+1}(A) = \alpha_{F-C}^S \cdot Q_t(A) + \lambda \cdot \alpha_L^S \cdot asset_t \cdot \Delta_t^S$, if $A$ is chosen. <br> $Q_{t+1}(A) = \alpha_{F-UC}^S \cdot Q_t(A)$, if $A$ is not chosen. <br> $\Delta_t^S = \Delta_G^S$, $\Delta_N^S$, $\Delta_L^S$ for gain, neutral, and loss at trial $t$. |

Building on the best model explaining normal behavior – the DF model, we asked how ketamine (0.5 mg/kg, IM) modulated the parameters of this model by testing variants of DF models, in each of which a different subset of parameters of the model were allowed to vary/deviate freely from the baseline parameters estimated from saline sessions (*Table 2*).

In all three animals, the model incorporating valence-dependent change in outcome evaluation best fit the choice data from ketamine sessions with (*K*-model 7 in the parenthesis, P) or without (*K*-model 4, P and Y/B) additional change in the tendency of choice perseveration (*Figure 4* and *Table 3*). Under ketamine, value update increased positively following all outcomes, thus increasing overall probability of repeating the same choice or namely, choice perseveration. However, the change was larger for non-gain outcomes, with the largest modulation after loss, consistent with the results from

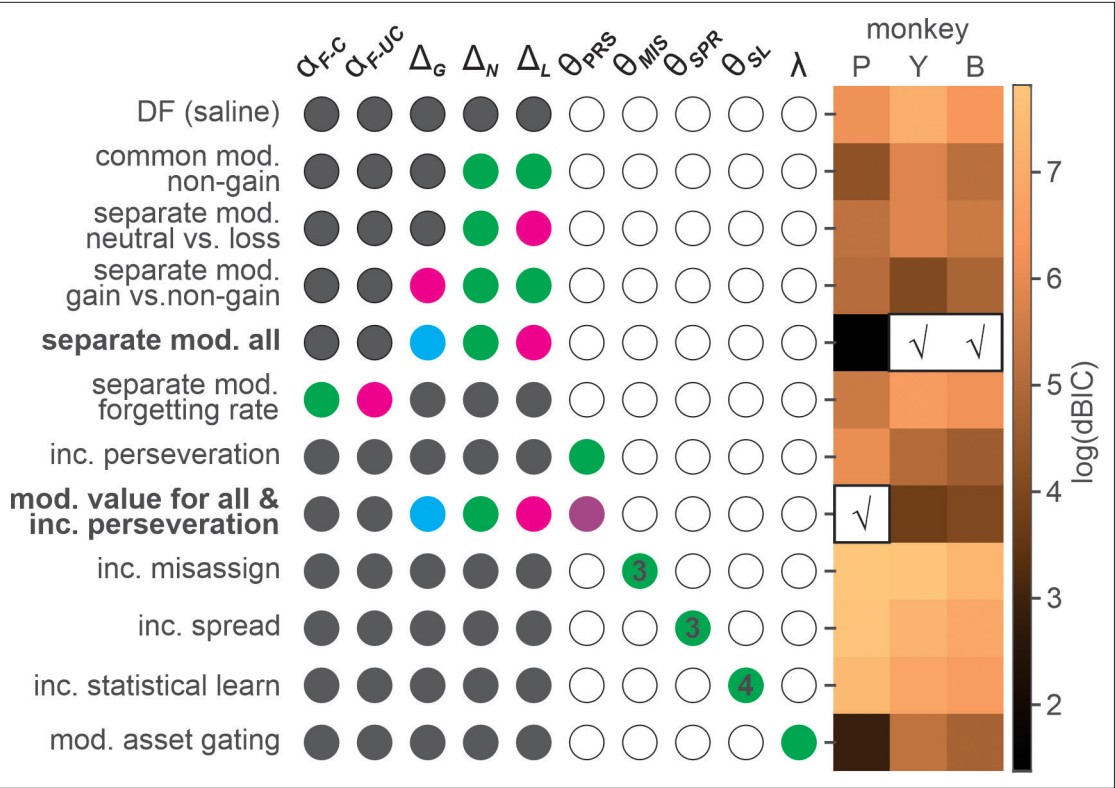

**Figure 4.** Model comparison for the effects of ketamine. Each row shows fixed (gray-filled circles), free (colored circles), and unused (empty circles) parameters of a variant of differential forgetting (DF) model. For the details of each model, see *Table 2*. *θ* represents a set of parameters added to a particular variant of DF model, and the number inside colored circles indicate the number of added parameters (>1). For the details of each model, see *Table 2*. Heat map on the right represents natural logarithm of differential BIC (Bayesian information criterion) of each variant from the best model. Best model for each animal is indicated by check (√) mark. *DF(saline)*: differential forgetting model fit to the data from saline session; *Common mod. non-gain*: common modulation of non-gain outcome evaluation; *Separate mod. neutral vs. loss*: differential modulation of neutral and loss outcome evaluation; *Separate mod. gain vs. non-gain*: differential modulation of gain and non-gain outcome evaluation; *Separate mod. all*: outcome-dependent modulation of outcome evaluation; *Separate mod. forgetting rate*: differential modulation of forgetting rate for chosen and unchosen target; *Inc perseveration*: increased perseveration; *Mod. value for all & inc perseveration*: outcome-dependent modulation of outcome evaluation and increased perseveration; *Inc misassign*: increased backward credit misassignment; *Inc spread*: increased forward spread in credit assignment; *Inc statistical learning*: increased statistical learning of reward rate; *Mod. asset gating*: modulation of asset-gated outcome evaluation.

The online version of this article includes the following source data for figure 4:

**Source data 1.** Numerical values of BIC used to generate the figure.

**Table 3.** Maximum likelihood parameter estimates of the best models for saline and ketamine sessions.

| | | $\alpha_{F-C}$ | $\alpha_{F-UC}$ | $\Delta_G$ | $\Delta_N$ | $\Delta_L$ |
|---|---|---|---|---|---|---|
| P | Saline | 0.72 | 0.09 | −0.57 | −1.3 | −2.31 |
| | Ketamine | | | 0.40 (0.49) | 0.11 (0.18) | −0.18 (−0.15) |
| Y | Saline | 0.65 | 0.20 | −0.54 | −1.91 | −3.15 |
| | Ketamine | | | 0.31 | −0.18 | −0.95 |
| B | Saline | 0.83 | 0.19 | −1.07 | −2.26 | −4.12 |
| | Ketamine | | | 0.18 | −0.25 | −1.01 |

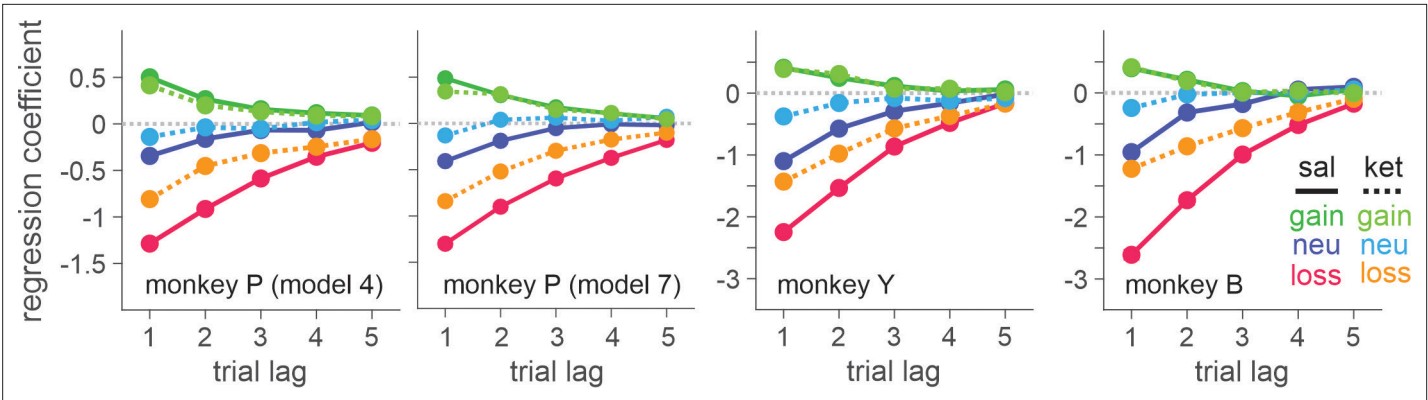

**Figure 5.** Ketamine-induced behavioral modulation simulated with differential forgetting model (for saline session) and best-fitting *K*-model (for ketamine session). Simulated data were generated with the maximum likelihood parameters of best-fitting models for saline (differential forgetting model) and ketamine (*K*-model 4: differential modulation of value for all the outcomes) sessions. Simulated choice was analyzed with logistic regression model (*Equation 1*).

The online version of this article includes the following source data and figure supplement(s) for figure 5:

**Source data 1.** Means and standard errors of regerssion coefficients.

**Figure supplement 1.** Ketamine-induced behavioral modulation simulated with best-fitting *K*-model of each class of reinforcement learning (RL) models.

**Figure supplement 1—source data 1.** Means and standard errors of regression coefficients.

our model comparison that the model including only a perseveration component performed worse than the best model in all three animals (*Figure 4*).

When the effect of ketamine was simulated with the best models within each class of ketamine models, *K*-model 4 (differential modulation of outcome evaluation for all three outcomes) produced the most similar results to actual data (*Figure 5*, *Figure 5—figure supplement 1*).

Unlike monkeys Y and B, the best-fitting model for monkey P indicated that ketamine reduced overall tendency to switch choice in addition to outcome-dependent modulation of outcome evaluation. However, Bayesian information criterion (BIC) differed only slightly (dBIC = 3.99) between the best-fitting (*K*-model 7) and the second-best model (*K*-model 4) and the model predictions for choice behavior were very similar both qualitatively and quantitatively (*Table 3* and *Figure 5*). We conclude that the behavioral effects of ketamine were consistent across all three monkeys. We did not find convincing evidence to support that ketamine significantly modulated cognitive (i.e. memory, temporal credit assignment [TCA] models) or motivational (i.e. motivation model) aspects of the animal's behavior. However, ketamineinduced ocular nystagmus particularly when animals were holding fixation (i.e. pre-feedback fixation) on a peripheral target immediately after a choice was made.

Centripetal eye drift measured by eye velocity was significantly larger under ketamine (*Figure 6*; 0.5 mg/kg IM, three-way interaction among drug, dose, and time in ANOVA, $F(3, 523) = 20.69$, $p < 0.001$ for P, $F(3,439) = 118.33$, $p < 0.001$ for Y, two-way interaction between drug and time, $F(3, 124) = 40.72$, $p < 0.001$ for B; 1 mg/kg IN, three-way interaction, $F(3, 103) = 3.54$, $p = 0.018$ for Y, $F(3,242) = 6.26$, $p < 0.001$ for B, main effect of drug, $F(1, 326) = 20.61$, $p < 0.001$ for P). Overall magnitude of nystagmus was significantly larger when ketamine was administered IM than IN (*t*-test, $p < 0.001$ for all three animals).

Ketamine's effects on both ocular nystagmus and negative evaluation of loss peaked and waned away within 1 hr after the injection (*Figures 6 and 7*). Despite the similar time course, further analysis suggested that these two effects were unlikely to be produced by a common mechanism/process. Nystagmus tended to cause fixation errors during the pre-feedback waiting period in a small number of trials. We found that as the animals owned more tokens and therefore were closer to earning juice reward, fixation errors became less frequent under ketamine (two-way interaction between drug condition and # of owned token in linear regression, $t = -3.6$, $-3.3$, and $-9.5$ for P, Y, and B, respectively, $p < 0.001$ for all three animals; *Figure 8A*), suggesting that animals could counter the

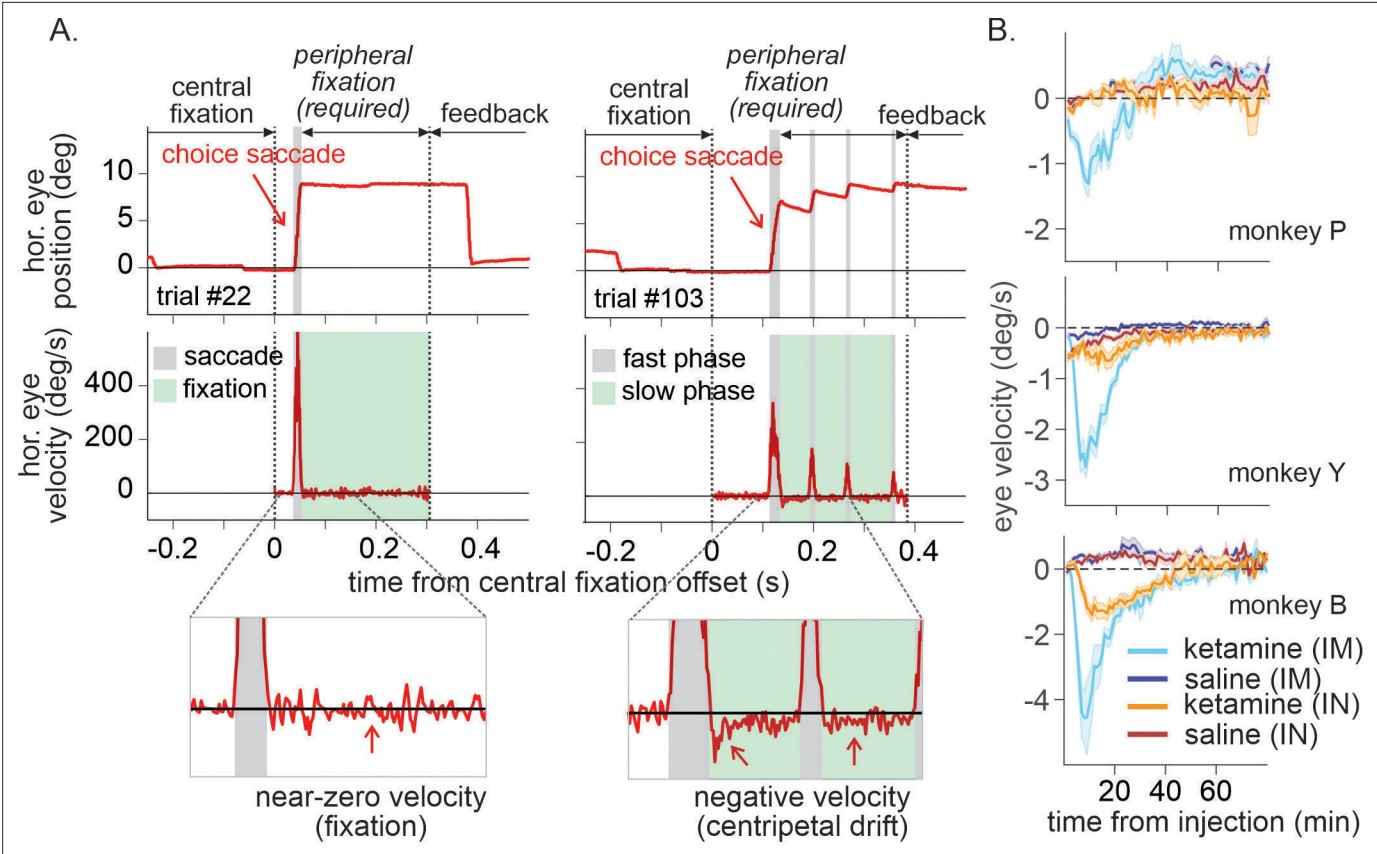

**Figure 6.** Time course of ketamine-induced ocular nystagmus. (**A**) Ocular position and velocity during fixation on the peripheral target in example trials during saline (left panel) and ketamine (right panel) sessions. (**B**) Time course of mean ocular velocity aligned at the time of saline or ketamine injection (monkey P, Y, and B from top to bottom). Shades indicate standard error. IM and IN indicates intramuscular and intranasal administration, respectively. For IM sessions, N=52 (Sal) and 17 (Ket) for monkey P; N=49 (Sal) and 15 (Ket) for monkey Y; N=25 (Sal) and 9 (Ket) for monkey B. For IN sessions, N=23 (Sal) and 13 (Ket) for monkey P; N=16 (Sal) and 10 (Ket) for monkey Y; N=19 (Sal) and 8 (Ket) for monkey B.

The online version of this article includes the following source data for figure 6:

**Source data 1.** Means and standard errors of slow-phase ocular velocity (deg/s) used to generate *Figure 6B*.

undesirable effect of ketamine when they had stronger motivation to obtain juice reward. However, ketamine-induced modulation of loss evaluation (i.e. reduced probability of choice switch after loss) did not systematically change with animals' motivation gated by the number of accrued tokens (two-way interaction between drug condition and # of owned tokens in linear regression, p > 0.25 for all three animals; *Figure 8B*).

## Discussion

The acute behavioral effects of ketamine that might reflect antidepressant activity have seldom been systematically examined as the potential antidepressant effects could be obscured by dissociative side effects in the assessment based on self-report/rating (*Berman et al., 2000*; *Sanacora et al., 2017*; *Gould et al., 2019*). Using a token-based decision task and extensive computational modeling, we examined the behavioral modulation induced by therapeutic doses of ketamine to gain insights into possible early signs of ketamine's antidepressant activity. We identified the robust effect of ketamine in attenuating negative evaluation of decision outcomes, without affecting the animal's motivation to obtain juice reward, the ability for proper TCA or the time constant of memory decay for decision outcomes.

Negative bias in perception, memory, feedback, and emotional processing has been reported to be one of the core behavioral/neuropsychological features of depression and negative affective bias, in particular, was proposed to play a causal role in the development, maintenance, and treatment

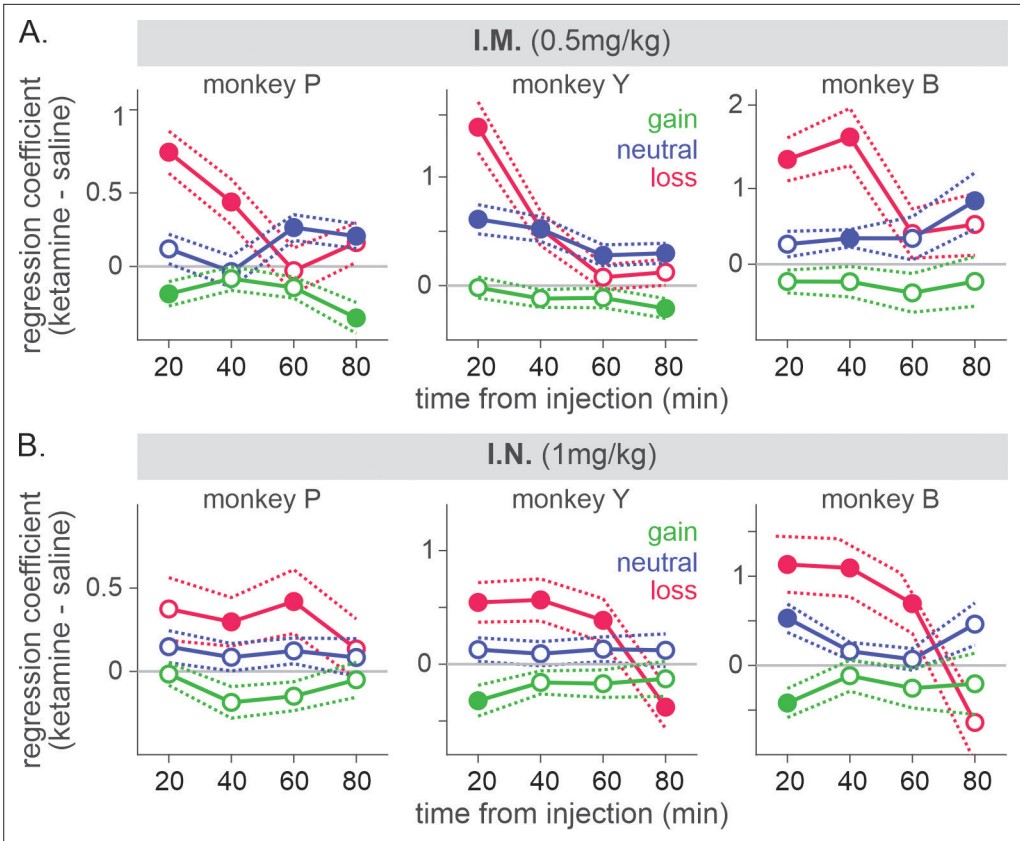

**Figure 7.** Time course of ketamine-induced modulation of outcome-dependent choice behavior. Time course of attenuation in loss evaluation induced by 0.5 mg/kg of intramuscularly (**A**) and 1 mg/kg of intranasally (**B**) administered ketamine. Regression coefficient reflecting ketamine's modulation of the effect of each outcome from the previous trial (trial lag 1) is plotted as a function of time relative to the injection. Solid (empty) symbols indicate that coefficients are (not) significantly different from zero (t-test, $p < 0.05$). Dotted lines represent standard error obtained from shuffled data (N=1000) between saline and ketamine sessions separately for each outcome. For IM sessions, N=52 (Sal) and 12 (Ket) for monkey P; N=49 (Sal) and 15 (Ket) for monkey Y; N=4 (Sal) and 7 (Ket) for monkey B.

The online version of this article includes the following source data for figure 7:

**Source data 1.** Means and standard errors of differential regression coefficients.

of depression (*Clark et al., 2009*; *Roiser et al., 2012*; *Godlewska and Harmer, 2021*). Changes in emotional processing – increased (decreased) positive (negative) affective bias early in the course of antidepressant treatment were also reported to predict later clinical response (*Tranter et al., 2009*; *Shiroma et al., 2014*; *Browning et al., 2019*; *Rychlik et al., 2017*). While ketamine's antidepressant effect is reported to be sustained over a week of period (*Zarate et al., 2006*), ketamine's effect on outcome evaluation was acute and did not last over subsequent days (*Figure 9*). This discrepancy might be attributable to the possible differences in the state of brain network between healthy subjects and those with depression as well as the type of measures taken to assess ketamine's effect. It is noteworthy that even in our task in which the contingency of gains and losses were designed to reinforce dynamic choice, the impact of loss was cumulative as memory decayed slowly over trials and therefore, attenuation of initial impact of loss had a longer-term effect beyond the time of initial evaluation (*Figure 1C*). In real-world situations where life events have consequences and their memory decays at much longer time scales than in our study, ketamine's acute/early effect might be able to mediate longer-term changes in negative affective processing/schemata and mood while patients continue to interact with their social environment during the course of treatment (*Godlewska and Harmer, 2021*; *Shiroma et al., 2014*; *Soltani et al., 2021*). Nevertheless, systematic studies are

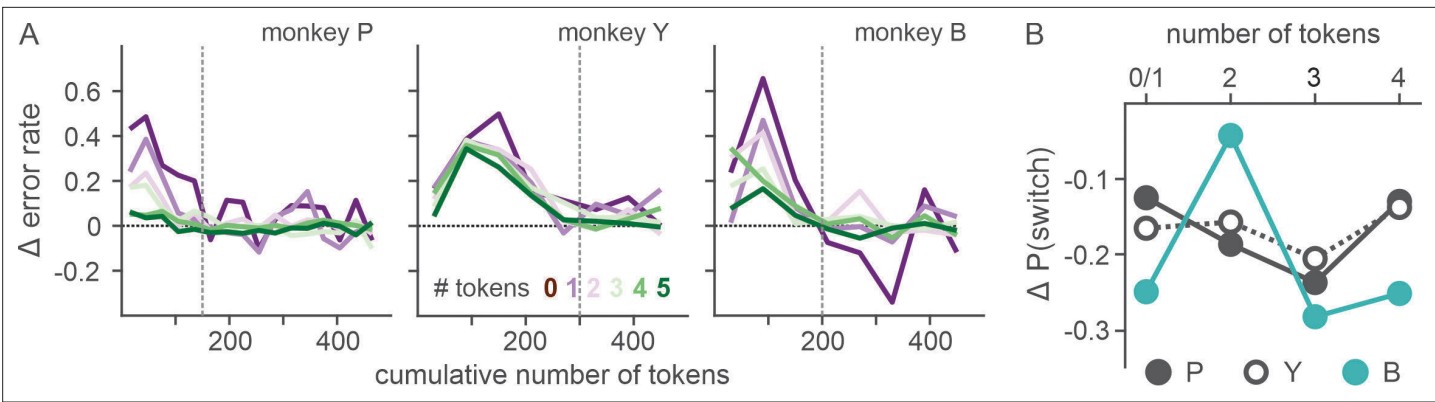

**Figure 8.** Effect of motivation on countering ketamine-induced fixation errors. (**A**) Difference in the rate of fixation break/trial between ketamine and saline sessions is plotted as a function of cumulative number of tokens (as a proxy for time with the satiation effect being controlled). Number of tokens owned by the animal at a given trial (asset) is color coded. Vertical dotted lines demarcate the latest data point that was included in the linear regression analysis. (**B**) Difference in the probability of choice switch after loss from the previous trial (trial lag 1) between ketamine and saline sessions is plotted as a function of asset at the time of decision. Due to limited number of loss trials, analysis was performed after dividing trials into four groups according to asset (0–1, 2, 3, 4). N=52 (Sal) and 17 (Ket) for monkey P; N=49 (Sal) and 15 (Ket) for monkey Y; N=25 (Sal) and 9 (Ket) for monkey B.

The online version of this article includes the following source data for figure 8:

**Source data 1.** Differential error rate and probability of choice switch as the function of owned tokens.

required to understand whether the reduced aversiveness to loss in our task might share the same mechanisms that underlie ketamine's antidepressant action.

Ketamine has been used as a pharmacological model of schizophrenia inducing cognitive deficits (*Krystal et al., 1994*; *Blackman et al., 2013*; *Fleming et al., 2022*). Ketamine was reported to impair cognitive functions such as working memory, context-dependent behavioral control, task-switching, rule learning, and reasoning (*Taffe et al., 2002*; *Condy et al., 2005*; *Skoblenick and Everling, 2012*; *Krystal et al., 2000*; *Brunamonti et al., 2014*). We tested multiple hypotheses regarding ketamine's possible effects on the time constant of choice and outcome memory, accurate association between an outcome and its causative choice (i.e. TCA) and behavioral perseveration. Ketamine increased the overall probability of repeated choice. However, since this tendency was outcome dependent, with the effect being significantly larger for non-gain outcomes, and largest for loss, this effect cannot be entirely explained by increased perseveration alone. These results are consistent with a previous report that ketamine did not affect the process of learning, but only affected the integration of information for decision (*Scholl et al., 2014*). These results also suggest that the memory of affective events and their associations (i.e. TCA) might be more resistant to acute disruption by NMDA receptor blockade, than affectively neutral information or rules. This disruption-resistant affective memory might underlie the delayed onset of antidepressant's therapeutic effect and treatment-resistant depression.

It was reported that oculomotor side effects induced by ketamine rapidly decreased over repeated administrations (*Pouget et al., 2010*). Interestingly, we found that ketamine-induced fixation errors could be countered at a very short time scale (i.e. on a trial basis) through animals' strong motivation to earn juice reward, further corroborating our findings that ketamine did not change the animal's motivation for reward.

Finally, our results demonstrate the sensitivity and the potential of the non-human primate model for investigation of the neurobiology of depression and antidepressant treatments, warranting further study on the mechanisms of ketamine's antidepressant action over different mood states and contexts carrying affective events that have impacts at diverse time scales.

## Materials and methods
### Animal preparation

Three male monkeys (*Macaca mulatta*, P, Y, and B; body weight, 12–16 kg) were used. The animal was seated in a primate chair with its head fixed, and eye position was monitored with a high-speed eye tracker (ET 49, Thomas Recording, Giessen, Germany). All procedures used in this study were

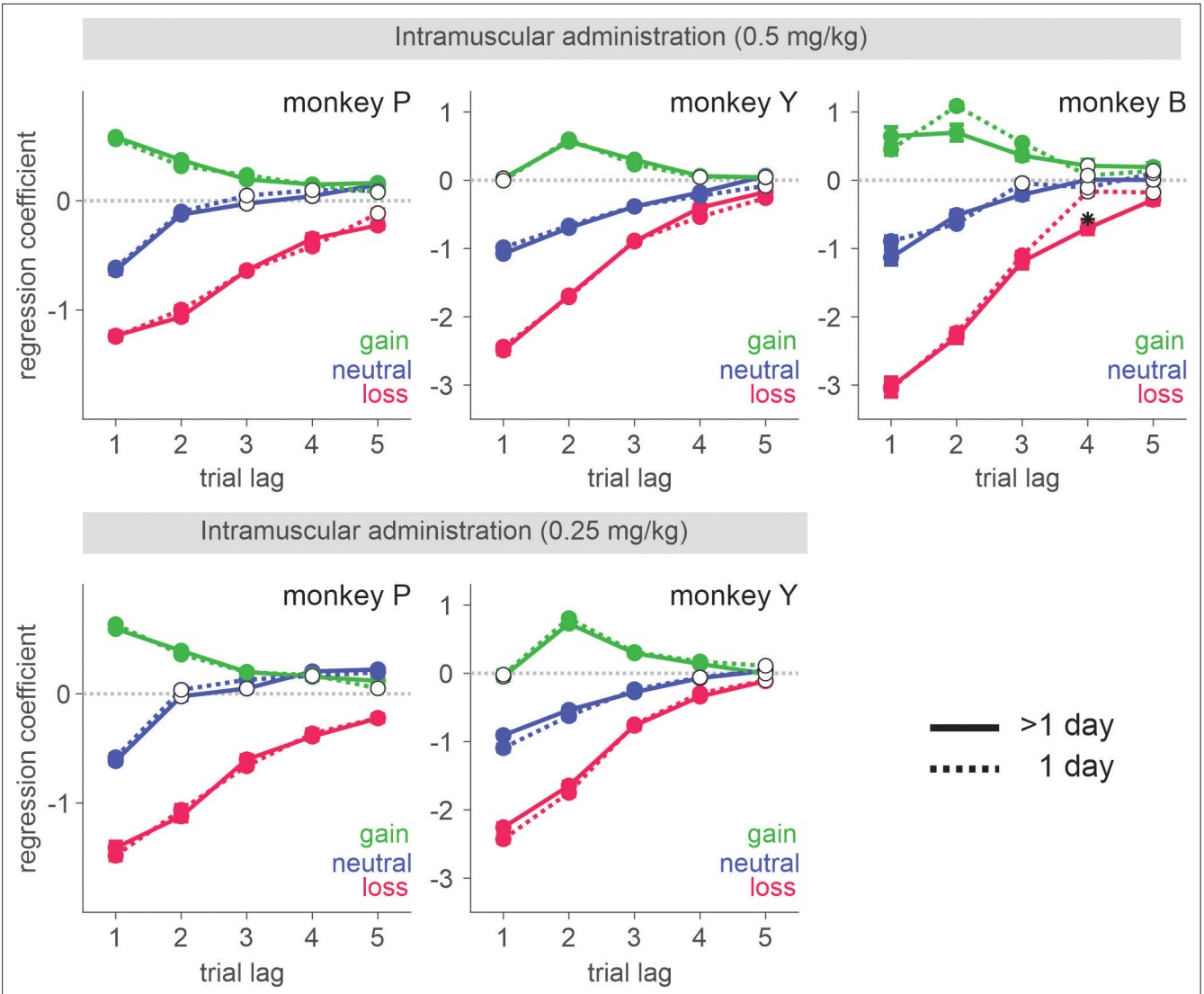

**Figure 9.** Behavioral effects of ketamine do not spread over the sessions subsequent to the injection. Regression coefficients reflecting the effects of gain, neutral (zero token), and loss outcomes obtained in the past trials are plotted. Solid (dotted) lines represent data from saline sessions >1 day (1 day) after a ketamine session. Solid (empty) symbols indicate that the corresponding coefficients are (not) significantly different from zero (t-test, $p <$ 0.05). Standard error is shown as horizontal bars above/below each coefficient. For IM sessions, N=37 (Sal) and 14 (Ket) for monkey P; N=35 (Sal) and 13 (Ket) for monkey Y; N=18 (Sal) and 6 (Ket) for monkey B. For IN sessions, N=37 (Sal) and 12 (Ket) for monkey P; N=25 (Sal) and 10 (Ket) for monkey Y.

The online version of this article includes the following source data for figure 9:

**Source data 1.** Means and standard errors of regression coefficients.

approved by the Institutional Animal Care and Use Committee at Yale University (Protocol #: 11058), and conformed to the Public Health Service *Policy on Humane Care and Use of Laboratory Animals* and *Guide for the Care and Use of Laboratory Animals*.

## Behavioral task

Animals were trained to perform a token-based BMP task (*Figure 1A*). Details of the task have been previously published (*Seo and Lee, 2009*). Briefly, animals played a competitive BMP game against a computerized opponent and the outcome of each trial was determined jointly by the animal's and the opponent's choice according to the payoff matrix (*Figure 1B*). The payoff matrix was adjusted for each animal to produce comparable choice behavior across all three animals. When the animals matched the opponent's choice, they gained one (monkey P) or two (monkey Y/B) tokens, whereas non-matching choices incurred either zero token (i.e. neutral outcome) or loss of one (Y/B) or two (P) tokens, with the

probability of loss being higher (lower) than that of zero token for the risky (safe) target. The locations of 'safe' and 'risky' targets were fixed during a block of 40 trials and then switched with a probability of 0.1 afterwards. The computer opponent simulated a rational player who exploits any predictable patterns in the animal's choice and outcome history to minimize the animal's payoff (*Seo and Lee, 2009*). To maximize payoff, the animals were required to dynamically change their choices to minimize serial correlations in the sequence of choice and outcome that could be exploited by the opponent.

## Administration of ketamine

For IM administration, 0.25 or 0.5 mg/kg of ketamine (Ketaset, Zoetis Inc) was injected into the right or left calf muscle. Doses were selected based on the therapeutic doses of ketamine when it is used as antidepressant for human patients. For IN administration, 0.5 or 1 mg/kg of ketamine was delivered into the right or left nostril using an IN mucosal atomization device (MAD Nasal, Teleflex Medical). Ketamine was administered with a 6.4-day interval on average (2–7 days), and sterile saline (IM) or sterile water (IN) were administered in-between successive ketamine sessions.

## Analysis of choice behavior

### Logistic regression model

The effect of past outcomes on an animal's choice was analyzed using the following model:

$$\text{logit}\left[P_t\left(R\right)\right] = \sum_{i=1}^{5} b_i^G G\left(t-i\right) + \sum_{i=1}^{5} b_i^N N\left(t-i\right) + \sum_{i=1}^{5} b_i^L L\left(t-i\right) + \sum_{i=1}^{5} b_i^R R\left(t-i\right) \tag{1}$$

where $P_t\left(R\right)$ is the probability of animal's choosing right target at the current trial $t$, $G(t-i)$, $N(t-i)$, $L(t-i)$, and $R(t-i)$ are regressors encoding gain, neutral, loss, and juice reward outcome that occurred after right (left) choice in the $i$th trial into the past as 1 ($-1$), and other outcomes as 0, respectively. $b_i^G, b_i^N, b_i^L, b_i^R$ are regression coefficients, with positive (negative) coefficients indicating that the particular outcome at trial $t-i$ tended to increase (decrease) the animal's tendency to choose the same target at trial $t$, relative to chance level (*Figure 1C*). Separate regression models were fit to saline, IM and IN administered ketamine sessions for each animal.

## Nonlinear regression – exponential model

We modeled the temporal decay of the effect of past outcome over multiple trials and its modulation by ketamine with the following exponential function:

$$Y_i^O = \left[b_0^s + b_0^k \cdot I_k\right] \cdot exp\left[-\left(i-1\right)/\left(b_1^s + b_1^k \cdot I_k\right)\right] \left(o = gain, neutral \vee loss\right) \tag{2}$$

where $i$ is trial lag ($i$th trial into the past), $I_k$ is an indicator variable encoding ketamine (saline) session as 1 (0). $b_0^s$ and $b_1^s$ are initial amplitude and decay time constant, respectively, for saline session, whereas $b_0^k$ and $b_1^k$ are modulation by ketamine. Exponential functions were fit to the data with nonlinear regression using *fitnlm.m* in *MATLAB* (MathWorks), separately for each outcome.

## RL models

We used RL models to investigate the potential cognitive/computational processes that might be modulated by ketamine. RL models have been extensively and successfully applied to explain reward-guided choice behavior (*Lee et al., 2012*). The general idea of RL is that the outcomes experienced from a particular action/state are integrated over time to form the reward expectation or the value of that action/state, and that an action among alternatives is chosen to maximize the expected outcome. One algorithm for the integration is to incrementally update the old expectation/value with a prediction error which can be formalized as follows for a single-state situation (*Sutton and Barto, 1998*):

$$Q_{t+1}\left(A\right) = Q_t\left(A\right) + \alpha_L \cdot \left[R_t - Q_t\left(A\right)\right] \left(standard Q - learning\right) \tag{3}$$

where $Q_t\left(A\right)$ is the value function for action $A$ at trial $t$, $\alpha_L$ is learning rate, and $R_t - Q_t\left(A\right)$ is the discrepancy between actual outcome, $R_t$ and the expectation/value, namely reward prediction error. This formula can be rearranged as follows *Barraclough et al., 2004*; *Ito and Doya, 2009*:

$$Q_{t+1}(A) = (1 - \alpha_L) \cdot Q_t(A) + \alpha_L \cdot R_t = \alpha_F \cdot Q_t(A) + \alpha_L \cdot R_t \qquad (4)$$

where $\alpha_F$ and $\alpha_L$ are separate parameters for the forgetting rate of the old value estimate and the learning rate of a new outcome, respectively. With $0 < \alpha_F < 1$, the old value function decays exponentially over time, simulating forgetting. Action selection was modeled with a softmax function,

$$P_t(R) = 1/\left(1 + e^{-\beta \cdot X_t}\right), X_t = [Q_t(R) - Q_t(L)] \qquad (5)$$

where probability of choosing right target, $P_t(R)$ was a logistic function of the estimated value of right relative to left choice. Inverse temperature, $\beta$ regulated randomness of choice determining the influence of action values on decision.

To investigate the effects of ketamine, we first tested multiple variants of the general model and determined the model that best explained normal behavior during saline sessions (*Table 1*; *Figure 3*). This is important, since we can expect to characterize the effect of ketamine accurately only in the context of a reliable behavioral model for how the animal's behavior was influenced by the outcomes of their previous choices.

Then, we examined how the parameters of this model were modulated by ketamine (*Table 2*; *Figure 4*).

## Model selection for saline sessions

In the most restricted, standard Q-learning model, value was updated only for the chosen action by taking a weighted average of $Q_t(A)$ and $R_t$ with $\alpha_F = 1 - \alpha_L$. $R_t$, was fixed as −1, 0, and 1 for gain, neutral, and loss outcome, respectively. These arrangements confined action value Q within the range of −1–1.

For three variants of RL models – Q-learning with subjective evaluation (QSE), DF, and non-differential forgetting (NDF) models, we relaxed the restrictions of the Q-learning model in two ways. *First*, $R_t$ was set as a free parameter reflecting subjective outcome evaluation. When both $\alpha_L$ and $R_t$ are set as free parameters, the model becomes underdetermined. Therefore, in these models, the product of learning rate and outcome value, $\alpha_L \cdot R_t$ was consolidated into a single parameter $\Delta_t$ or the total amount of change due to a new outcome which could take different values for gain ($\Delta_G$), neutral ($\Delta_N$), and loss ($\Delta_L$) outcome. We also tested the models in which $\Delta_N$ was fixed as 0 simulating effectively neutral outcome. *Second*, action value Q for the unchosen action was allowed to decay, with DF rates for chosen ($\alpha_{F-C}$) and unchosen ($\alpha_{F-UC}$) actions in a DF model, and with a common forgetting rate, $\alpha_F$ in an NDF model. Action values were updated only for the chosen action in a QSE model.

Finally, we tested an additional model in which outcome evaluation was gated by asset (non-DF with asset-gated outcome evaluation – NDF-A model) or the number of tokens owned by the animal at the time of decision, simulating temporal discounting of outcome value depending on its temporal distance to the primary reward that was always given only after six tokens were acquired.

For all tested models other than Q-learning model, $\beta$ was fixed to be 1, as a model was underdetermined when both $\beta$ and $\Delta_t$ could vary freely. Model parameters were fit to choice data to maximize the likelihood of data. BIC was used for model comparison and selection of the best model.

## Model selection for ketamine sessions

To understand the effect of ketamine, we started with the best-fitting model of normal behavior during saline sessions – the DF model, and then asked how ketamine might modulate the parameters of this model. The null (baseline) model assumed that ketamine did not cause any parametric changes in the behavior, and the model prediction of trial-to-trial choice was generated with the maximum likelihood (ML) parameters, namely 'baseline parameters' estimated from saline sessions as follows:

$$Q_{t+1}(A) = \alpha_{F-C}^S \cdot Q_t(A) + \Delta_t, \text{ if } A \text{ is chosen}, \qquad (6)$$

$$Q_{t+1}(A) = \alpha_{F-C}^S \cdot Q_t(A) + \Delta_t, \text{ if } A \text{ is not chosen}, \qquad (7)$$

$$\Delta_t = \Delta_G^S, \Delta_N^S, \Delta_L^S, \text{ for gain, neutral, and loss, respectively}.$$

$$P_t(R) = 1/\left(1 + e^{-X_t^S}\right), X_t^S = [Q_t(R) - Q_t(L)] \qquad (8)$$

where $\alpha_{FC}^S$ , $\alpha_{FUC}^S$ , $\Delta_G^S$ , $\Delta_N^S$ , $\Delta_L^S$ (baseline parameter-estimates) were ML parameters of the DF model fit to the data from saline sessions.

In each of the ketamine models, we set a subset of parameters of the DF model to freely vary from the baseline parameter-estimates. *Value models* (*K*-model 1–4, *Table 2*) hypothesized that ketamine changed outcome evaluation differentially for gain, neutral, and loss outcomes, whereas a *memory model* (*K*-model 5) tested whether ketamine affected the forgetting rate or the rate of value decay. *Perseveration models* hypothesized that ketamine affected the behavioral tendency to perseverate (*K*-model 6), possibly in tandem with a change in value update (*K*-model 7). *TCA models* tested whether ketamine changed the way a particular outcome properly updated the value of causative action, possibly leading to misassignment of the credit for an outcome to the choice that happened in the past (*K*-model 8), spread of the credit to future choice (*K*-model 9), or assignment of the credit for recent outcomes collectively to the frequently chosen target in the near past (*K*-model 10) (*Walton et al., 2010*; *Jocham et al., 2016*). Finally, a *motivation model* (*K*-model 11) hypothesized that ketamine changed the gain with which asset position modulated the value update for each outcome.

## Analysis of ketamine-induced ocular nystagmus

Ocular nystagmus during fixation on a peripheral target consists of two components – slow centripetal drift (i.e. slow phase), and a subsequent corrective saccade to re-capture the target (i.e. fast phase) (*Barnes, 1982*). Eye position was sampled and recorded at 250 Hz. We used average velocity of eye movement during pre-feedback fixation on a chosen target to detect the slow phase of the nystagmus induced by ketamine. The sign of eye velocity was adjusted in the way that centripetal eye movement during fixation had negative velocity. Fast phase eye movement with velocity >20° (visual angle)/s was first removed from the pre-feedback fixation period, and then eye velocity was averaged during the remaining time points of the fixation period for each trial (for statistical analysis) or in 1-min non-overlapping bins for visualization of time course (*Figure 6*).

## Analysis of plasma concentration of ketamine

To be able to effectively compare behavioral effects observed following IM and IN administration of ketamine, we measured plasma concentration of ketamine following IM and IN administrations in monkey B (one session for 0.5 mg/kg IM; two sessions for 0.5 mg/kg IN; two sessions for 1 mg/kg IN administration). Blood samples were collected at 10, 25, 40, 60, and 75 min following ketamine administration and analyzed with the high performance liquid chromatography (HPLC) protocol. 100 ng of the internal standard clonidine and 0.5 ml of thawed plasma were added to a 15-ml polypropylene tube and 15 ml of diethyl ether added. Following 5 min of vortex mixing, 4 ml of the upper ether layer was transferred to a new polypropylene tube and 250 µl of 0.01 M phosphoric acid added. After vortex mixing for 5 min, the lower aqueous layer was transferred to a 1.5-ml tube and evaporated under vacuum (~1 hr, SpeedVac evaporator, medium heat). The dry residue was dissolved in 150 µl of 0.01 sodium monobasic dihydrogen phosphate. 50 µl of the extract was injected and separated on a 3-µm Cyano Spherisorb HPLC column (4.6 × 100 mm) eluted with 80% 0.01 sodium acetate/20% acetonitrile (0.8 ml/min, 40°C). Norketamine, ketamine, and clonidine were detected by UV absorption (220 nm; Hitachi L-7400) with retention times of 6.1, 8.1, and 11.0 min, respectively; and with absolute detection limits of 0.2, 0.5, and 0.3 ng, respectively. Extracted standards were prepared using 0.5 ml aliquots of pooled human plasma. Plasma drug concentrations of samples were calculated knowing the peak height ratios of the extracted standards, the peak height ratios in samples and the amount of internal standard added (200 ng/ml). When injecting 50 µl of the extract, norketamine and ketamine were determined with concentration detection limits of 1 and 2 ng/ml, respectively.

## Acknowledgements

We thank Melissa Boucher and Khalil AbedRabbo for their technical assistance. We also thank Drs. Daeyeol Lee and Amy Arnsten for their helpful discussion and thoughtful comments throughout the study. This work was supported by the National Institutes of Health (R01 MH108643).

## Additional information

### Funding

| Funder | Grant reference number | Author |
|--------|------------------------|--------|
| National Institute of Mental Health | R01 MH108643 | Hyojung Seo<br>Mariann Oemisch |

The funders had no role in study design, data collection, and interpretation, or the decision to submit the work for publication.

### Author contributions

Mariann Oemisch, Data curation, Formal analysis, Investigation, Visualization, Methodology, Writing - original draft; Hyojung Seo, Conceptualization, Resources, Supervision, Investigation, Methodology, Project administration, Writing - review and editing

### Author ORCIDs

Hyojung Seo http://orcid.org/0000-0002-2928-9018

### Ethics

All procedures used in this study were approved by the Institutional Animal Care and Use Committee at Yale University, and conformed to the Public Health Service Policy on Humane Care and Use of Laboratory Animals and Guide for the Care and Use of Laboratory Animals (Protocol #: 11058).

Reviewer #1 (Public review): https://doi.org/10.7554/eLife.87529.3.sa1
Reviewer #2 (Public review): https://doi.org/10.7554/eLife.87529.3.sa2
Author response https://doi.org/10.7554/eLife.87529.3.sa3

## Additional files

### Supplementary files

• MDAR checklist

### Data availability

Supporting files have been provided containing the numerical data used to generate the figures.

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
