## [Editor Report · eLife assessment]

The authors use reinforcement learning modeling to study the alterations following acute ketamine in macaques. The evidence supporting the conclusion that ketamine reduces the impact of losses vs. neutral/gains is **solid**. In this version of this **valuable** study, the authors make more measured interpretations about the relationship between the processing of losses and ketamine's antidepressant effects.

---

## [Referee Report · Reviewer #1 (Public review)]

Oemisch and Seo use sophisticated reinforcement learning (RL) modeling to show that acute ketamine reduces the strength impact of losses vs neutral/gains on the subsequent trial performance of a token-based biased matching-pennies task. In this version, the authors make more measured interpretations about the potential relevance of their results to ketamine's antidepressant effects for the most part.

My prior review emphasized what I considered to be an over-interpretation of the relevance of their data (that I find interesting and of value) to mechanisms of action of ketamine's antidepressant effects. The authors have corrected those excesses exception for the last sentence of the introduction, which continues to suggest they are studying both mechanisms of antidepressant actions as well as the pathophysiology of depression.

---

## [Referee Report · Reviewer #2 (Public review)]

Oemisch and Seo set out to examine the effects of low-dose ketamine on reinforcement learning, with the idea that alterations in reinforcement learning and/or motivation might inform our understanding of what alterations co-occur with potential antidepressant effects. Macaques performed a reinforced/punished matching pennies task while under effects of saline or ketamine administration and the data were fit to a series of reinforcement learning models to determine which model described behavior under saline most closely and then what parameters of this best-fitting model were altered by ketamine. They found a mixed effect, with two out of three macaques primarily exhibiting an effect of ketamine on the processing of losses and one out of three macaques exhibiting an effect of ketamine on processing losses and perseveration. They found that these effects of ketamine appeared to be dissociable from the nystagmus effects of the ketamine.

The findings are novel, and the data suggesting that ketamine primarily affects on the processing of losses (under the procedures used) are solid. However, it is unclear whether the connection between the processing of losses and the antidepressant effects of ketamine is justified, and the current findings may be more useful for those studying reinforcement learning than those studying depression and antidepressant effects. In addition, the co-occurrence of different behavioral procedures with different patterns of ketamine effects, with one macaque tested with different parameters than the other two exhibiting effects of ketamine that were best fit with a different model than the other two macaques, suggests that there may be difficulty in generalizing these findings to reinforcement learning more generally.

(1) First, the authors should be more explicit and careful in the connection they are trying to make about the link between loss processing and depression. The authors call their effect a "robust antidepressant-like behavioral effect." However, there are no references to support this or discussion of how the altered loss processing would relate directly to the antidepressant effects. A few statements about a link to antidepressant effects have been removed or moderated, but many remain, including those in the abstract. The authors provide little to no support for this link, so the current version represents solid evidence for an effect on loss processing and incomplete or weak evidence for an antidepressant effect.

(2) It appears that the monkey P was given smaller rewards and punishers than the other two monkeys, and this monkey had an effect of ketamine on perseveration that was not observed in the other two monkeys. This may be due to this monkey being trained and tested before the other animals, but it does raise the issue of the generality of the authors' findings. It seems possible that the procedures used for the other two monkeys (with no deviation at all) might support the best-fit model that the authors favor. However, if changes in the size of the rewards and punishments suddenly make ketamine affect perseveration, then it suggests that ketamine's effect is highly parameter-specific. For example, might there be some parameters where ketamine would only alter perseveration and not loss processing?

---

## [Author Response]

The following is the authors’ response to the original reviews.

**Reviewer #1 (Public Review):**
The modeling approaches are very sophisticated, and clearly demonstrate the selective nature of acute ketamine to reduce the impact of trial losses on subsequent performance, relative to neutral or gain outcomes. The authors then, not unreasonably, suggest that this effect is important in the context of the negative bias in interpreting events that is prominent in depression, in that if ketamine reduces the ability of negative outcomes to alter behavior, this may be a mechanism for its rapid acting antidepressant effects.However, there is a very strong assumption in this regard, as shown by the first sentence of the discussion which implies this is a systematic study of ketamine's acute antidepressant effects. In actuality, this is a study of the acute effects of ketamine on reinforcement learning (RL) modeled parameters. A primary concern here is that an effect presented as a "robust antidepressant-like behavioral effect" should be more enduring than just an alteration during the acute administration. As it is, the link to an "anti-depressant effect" is based solely on the selective effects on losses. This is not to say this is not an interesting observation, worthy of exploration. It is noted that a similar lack of enduring effects on outcome evaluation is observed in humans, as shown in supplemental fig. S4, but there is not accompanying citation for the human work.

We agree with the reviewer that the way we linked the study results to ketamine’s antidepressant action can be misleading and based on a rather strong assumption which was not systematically tested in the study. We made the following changes to the manuscript:

(1) These results constitute a rare report of a robust antidepressant-like behavioral effect produced by therapeutic doses of ketamine during acute phase (<1 hour) after injection (Introduction, 3rd paragraph, line 8-9 in the original manuscript).

Changed to: These results constitute a rare report of an acute effect of therapeutic dose of ketamine on the processing of affectively negative events during dynamic decision-making.

(2) We clarified in the Discussion that our study is to gain insights into, but not a systematic investigation of ketamine’s antidepressant action as follows:

(2.1) A sentence was added (1st paragraph of Discussion): Using a token-based decision task and extensive computational modeling, we examined the behavioral modulation induced by therapeutic doses of ketamine to gain insights into possible early signs of ketamine’s antidepressant activity.

(2.2) Consistent with the findings from humans, ketamine’s effect on outcome evaluation was acute and did not last over subsequent days (Supplemental Figure S4) (Discussion, 2nd paragraph, line 6-7 in the original manuscript).

Changed to: While ketamine’s antidepressant effect is reported to be sustained over a week of period (5), ketamine’s effect on outcome evaluation was acute and did not last over subsequent days (Supplemental Figure S4). This discrepancy might be attributable to the possible differences in the state of brain network between healthy subjects and those with depression as well as the type of measures taken to assess ketamine’s effect.

(2.3) A sentence was added (Discussion, last sentence of the 2nd paragraph) : Nevertheless, systematic studies are required to understand whether the reduced aversiveness to loss in our task might share the same mechanisms that underlie ketamine’s antidepressant action.

One question that comes to mind in terms of the selectivity observed is whether similar work has been done to examine the acute effects of any other drugs. If ketamine is unique in this regard, that would be quite interesting.

We think this is an interesting idea. However, comparing ketamine’s effect to that of other drugs is not the scope of the current study. We hope that we will be able to answer this question with future studies.

**Reviewer #2 (Public Review):**
Oemisch and Seo set out to examine the effects of low-dose ketamine on reinforcement learning, with the idea that alterations in reinforcement learning and/or motivation might inform our understanding of what alterations co-occur with potential antidepressant effects. Macaques performed a reinforced/punished matching pennies task while under effects of saline or ketamine administration and the data were fit to a series of reinforcement learning models to determine which model described behavior under saline most closely and then what parameters of this best-fitting model were altered by ketamine. They found a mixed effect, with two out of three macaques primarily exhibiting an effect of ketamine on processing of losses and one out of three macaques exhibiting an effect of ketamine on processing of losses and perseveration. They found that these effects of ketamine appeared to be dissociable from the nystagmus effects of the ketamine.The findings are novel and the data suggesting that ketamine is primarily having its effects on processing of losses (under the procedures used) are solid. However, it is unclear whether the connection between processing of losses and the antidepressant effects of ketamine is justified and the current findings may be more useful for those studying reinforcement learning than those studying depression and antidepressant effects. In addition, the co-occurrence of different behavioral procedures with different patterns of ketamine effects, with one macaque tested with different parameters than the other two exhibiting effects of ketamine that were best fit with a different model than the other two macaques, suggests that there may be difficulty in generalizing these findings to reinforcement learning more generally.(1) First, the authors should be more explicit and careful in the connection they are trying to make about the link between loss processing and depression. The authors call their effect a "robust antidepressant-like behavioral effect" but there are no references to support this or discussion of how the altered loss processing would relate directly to the antidepressant effects.

We agree with the reviewer’s point on the way we made the connection between the study results and ketamine’s antidepressant action. This concern overlaps with the reviewer #1’s concern. Please refer to our response 2, 2-1, 2-2 and 2-3.

(2) It appears that the monkey P was given smaller rewards and punishers than the other two monkeys and this monkey had an effect of ketamine on perseveration that was not observed in the other two monkeys. Is this believed to be due to the different task, or was this animal given a different task because of some behavioral differences that preceded the experiment? The authors should also discuss what these differences may mean for the generality of their findings. For example, might there be some set of parameters where ketamine would only alter perseveration and not processing of losses?

Although the best-fitting ketamine model for monkey P includes an additional element – perseveration, we believe that monkey P’s baseline behavior and ketamine’s effect are not significantly different from the other two monkeys for the following reasons.

First, monkey P was the first animal that we tested ketamine’s effect, and therefore we aimed to match the other two monkeys’ baseline behavior similar to monkey P’s behavior in order to reduce variability in ketamine’s effect potentially attributable to the difference in baseline behavior before pharmacological manipulation. We had to adjust the payoff matrix for the subsequent animals (Y and B) because these monkeys were more sensitive to loss, and seldom chose “risky” target (yielding loss). In order to make the other two monkeys’ behavior similar to that of monkey P, we adjusted the asymmetry between the risky and the safe target in the way that loss (neutral) outcome occurred from the safe (risky) target as well. Eventually, this adjustment made the baseline behavior similar across all three monkeys. The goal of the study was to reliably measure the ketamine’s effect, and not to study individual differences that can naturally occur with the same task parameters. Therefore, we believe that the adjustment of payoff matrix helped to reliably detect ketamine’s effect starting from the common baseline behavior.

Second, the best-fitting model for monkey P (K-model 7) and that for the other two monkeys (K-model 4) make very similar predictions both qualitatively and quantitatively as are seen in the revised Figure 4. The parameters for outcome values estimated from these two models in monkey P are very similar as is seen in the revised Table 3. In addition, the difference in BIC between the model which includes only perseveration modulation (K-model 6) and the model incorporating outcome value modulation as well (K-model 7) is 441, whereas the difference in BIC between K-model 7 and the model that includes only outcome value modulation (K-model 4) is as small as 4. These BIC results indicate that the variability explained by ketamine’s modulation of outcome evaluation is remarkably larger that that explained by its modulation of perseveration in monkey P.

Therefore, we conclude that ketamine’s effect was not significantly different between monkey P and the other two monkeys. We clarified this in the revised manuscript by adding the following paragraph in the Result section:

“Unlike monkey Y and B, the best-fitting model for monkey P indicated that ketamine increased overall tendency to switch choice in addition to outcome-dependent modulation of outcome evaluation. However, BIC differed only slightly (dBIC = 3.99) between the best-fitting (K-model 7) and the second-best model (K-model 4) and the model predictions for choice behavior were very similar both qualitatively and quantitatively (Table 3, Figure 4). We conclude that the behavioral effects of ketamine were consistent across all three monkeys.”

(3) The authors should discuss whether the plasma ketamine levels they observed are similar to those seen with rapid antidepressant ketamine or are higher or lower.

We added a sentence in the first paragraph of the Result section as follows with a reference.

“Plasma concentration and its time course over 60 minutes were also comparable to those measured after 0.5mg/kg in human subjects (35).”

(35) Zarate CA, Brutsche N, Laje G, Luckenbaugh DA, Venkata SLV, Ramamoorthy A, et al (2012): Relationship of ketamine’s plasma metabolites with response, diagnosis, and side effects in major depression. Biol Psychiatry, 72: 331-338.

(4) For Figure 4 or S3, the authors should show the data fitted to model 7, which was the best for one of the animals.

We added the parameters and model predictions from both K-model 7 and K-model 4 for monkey P to help comparison between two models in Table 3, and Figure 4. Revised Table 3 and Figure 4 are as follows:

**Author response table 1. sa3table1:** Maximum likelihood parameter estimates of the best models for saline and ketamine sessions.

		αF−C	αF−UC	ΔG	ΔN	ΔL
P	Saline	0.72	0.09	–0.57	–1.3	–2.31
	Ketamine			0.40 (0.49)	0.11 (0.18)	–0.18 (-0.15)
Y	Saline	0.65	0.20	–0.54	–1.91	–3.15
	Ketamine			0.31	–0.18	–0.95
B	Saline	0.83	0.19	–1.07	–2.26	–4.12
	Ketamine			0.18	–0.25	–1.01

In all three animals, the model incorporating valence-dependent change in outcome evaluation best fit the choice data from ketamine sessions with (K-model 7 in the parenthesis, P) or without (K-model 4, P and Y/B) additional change in the tendency of choice perseveration (Figure 3, Table 3).

**Author response image 1. sa3fig1:** ketamine-induced behavioral modulation simulated with differential forgetting model (for saline session) and best-fitting K-model (for ketamine session).